## Research Article

Sea-level rise; Coastal floods; Coastal Environments; Coastal community exposure; Extreme sea-level events

**Corresponding author:**
Bjorn Nyberg;
Email: bjorn.nyberg@uib.no

# Global coastal exposure patterns by coastal type from 1950 to 2050

Björn Nyberg[2,3,4], Albina Gilmullina[2], William Helland-Hansen[2,3], Jaap Nienhuis[1] and Joep Storms[5]

[1]Department of Physical Geography, Utrecht University, Utrecht, The Netherlands; [2]Department of Earth Sciences, University of Bergen, Bergen, Norway; [3]Climate Hazards Group, Bjerknes Centre for Climate Research, Bergen, Norway; [4]Innovation Centre, 7Analytics, Bergen Norway and [5]Department of Geoscience and Engineering, Delft University of Technology, Delft, The Netherlands

## Abstract

Addressing sea-level rise and coastal flooding requires adaptation strategies tailored to specific coastal environments. However, a lack of detailed geomorphological data on global coasts impedes effective strategy development. This research maps seven coastal environments worldwide, and for each environment analyzes the effect of coastal changes on coastal populations by including sea-level change, extreme sea-level events with varying return periods and population growth from 1950 to 2050. It identifies the historical exposure of low-lying deltaic and estuarine flood areas (>48% of total population) and reveals that flood exposure will significantly increase for barrier islands and strandplains by 2050 (with over a 40% rise in exposure), particularly along African coastlines. Population growth emerges as the primary factor behind the increased exposure. While sea-level rise is projected to contribute between 26% and 65% of the increased inundated area by 2050 compared to a 10-year extreme sea-level event, varying by coastal environment. The findings highlight the critical need for mitigation measures that account for the distinct responses of different coastal types to sea-level rise, posing various risks over varying timescales.

## Impact statement

This research underscores the pressing need for environment-specific strategies to address the growing threats of sea-level rise and coastal flooding. By mapping and analyzing seven distinct coastal environments worldwide, it identifies critical exposure patterns essential for shaping effective adaptation strategies and guiding future research priorities. The findings reveal that historically flood-prone, low-lying regions, such as deltaic and estuarine areas, continue to host over 48% of at-risk populations. However, coastal communities on barrier islands and strandplains face rapidly increasing risks, with exposure projected to rise by over 40% by 2050, particularly in African regions experiencing significant population growth. By highlighting how flood exposure drivers differ across coastal environments, the study emphasizes that one-size-fits-all mitigation approaches are inadequate. Instead, tailored strategies are necessary to protect vulnerable populations and enhance resilience. This work calls for adaptive planning that addresses the unique and evolving risks of diverse coastal landscapes, ensuring effective protection against this long-term impact of climate change.

## Introduction

Coasts and coastal lowlands are of great societal, economical and agricultural value with IPCC estimating nearly 40% of the world's population lives within 100 km of the coast (IPCC, 2023). With the expected acceleration of sea-level rise (Kulp and Strauss, 2019; Fox-Kemper, 2021) and intensification of storm surge events (Muis et al., 2020) throughout this century, understanding future coastal change is important to assess and mitigate projected coastal inundations. Furthermore, the consequences of extreme sea-level events for population, infrastructure and livelihoods, will vary depending on the type of coast (Passeri et al., 2015). Understanding the type of coastal environment is thus a critical component to assess future coastal inundation risks.

Current estimates on the impact of rising sea levels and coastal inundations predict that by the year 2100, up to an estimated 630 million people will be at risk given an RCP 8.5 emission scenario and an Antarctic instability (Kulp and Strauss, 2019). Yet there remains a significant knowledge gap on how the different types of coastal environments respond to sea-level rise and how these different environments condition the impact on the affected coastal communities. Several existing studies have focused solely on coastal communities on deltas due to their low-lying geographical positions with significant settlements and high agricultural productivity (Syvitski

et al., 2009; Nienhuis et al., 2023). Deltas are undoubtedly important in this respect; however, there are a range of additional coastal environments, such as rocky coasts, strandplains, barrier islands and estuaries (Figure 1), each of them responding distinctively to sea-level

change. The lack of a coherent and systematic assessment of global coastal types (GCTs) limits our ability to target the effects of sea-level rise and storm surge events across the range of different coastal environments (Hinkel et al., 2013). Risk assessment concerning

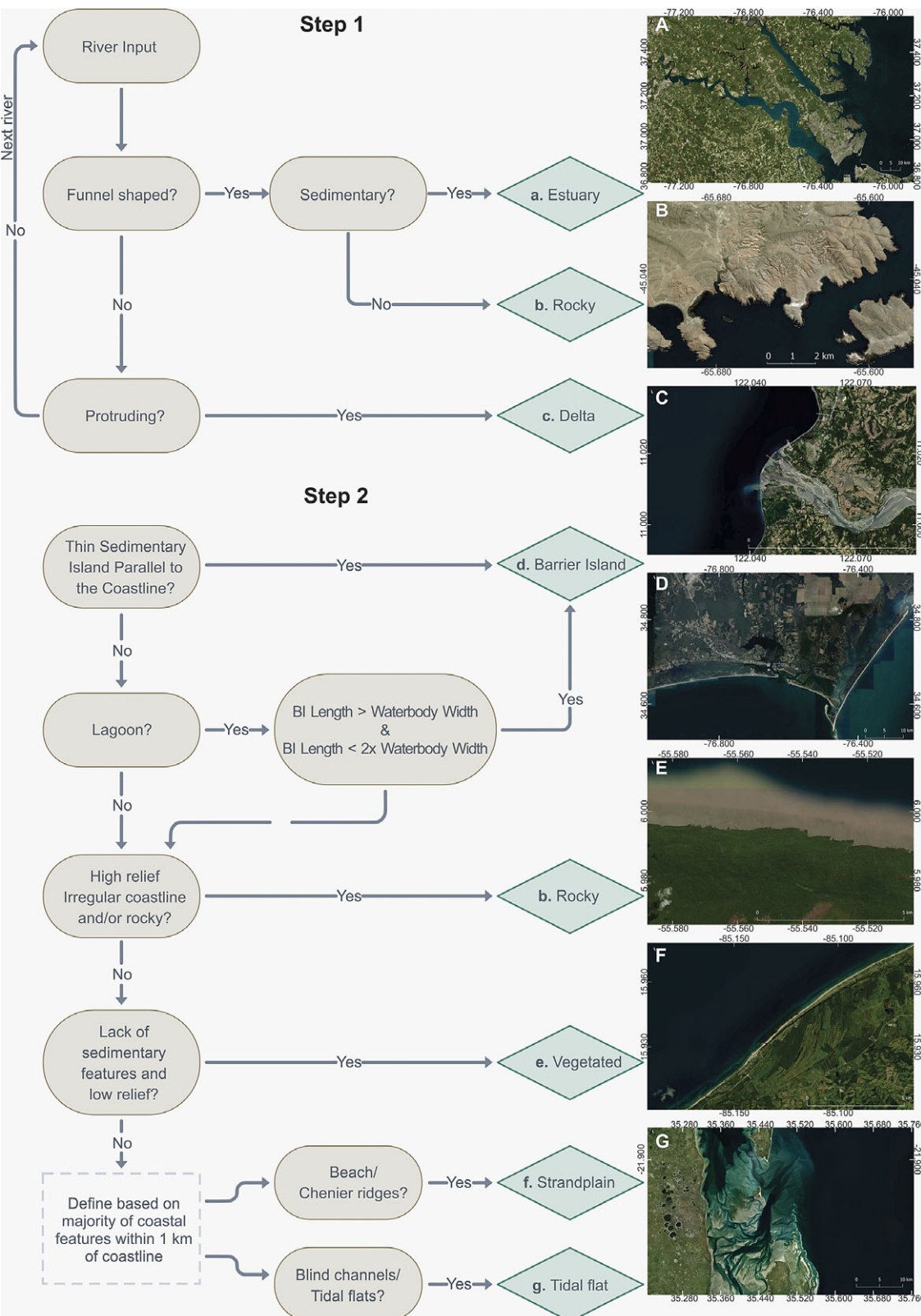

**Figure 1.** Flow chart showing the two-step workflow used to classify each coastal type based on key landform characteristics. Images are derived from Bing(c).

future flood inundations is inevitably different for a rocky coastline compared to a coast dominated by barrier islands.

Dürr et al. (2011) developed one of the first global typologies of coastal systems for estuaries. However, their study relied on hydrological catchment delineations that do not fully capture the coastal variability along shorelines. More recent research has shifted toward global assessment of specific coastal environments, such as beaches (Luijendijk et al., 2018; Vousdoukas et al., 2020), wetlands (Hu et al., 2017; Davidson et al., 2018), barrier islands (Mulhern et al., 2017) or deltas (Caldwell et al., 2019). While these "single-environment" studies provide valuable insights, they do not compare different coastal environments, making it difficult to assess how coastal typologies influence broader coastal processes. Only in the last few years has research focused on broader global coastal classifications. For example, Mao et al. (2022) applied a machine learning approach to delineate global coastlines into three broad categories: rocky, wetland or beach. Athanasiou et al. (2024) also present a classification of global coastlines based on physical characteristics, such as topography, bathymetry, land cover and vegetation type. Others have considered the oceanographic forces (e.g., tides and waves) influencing the shape of coasts, without consideration of the breakdown into specific coastal environments (Nyberg and Howell, 2016; Vulis et al., 2023). Finally, at finer spatial scales, regional databases like the Mediterranean Coastal Database have demonstrated the importance of physical, ecological and socioeconomic factors in assessing and driving policy to mitigate sea-level rise risks (Wolff et al., 2018).

Here, we map and characterize the different types of coastal environments to define a new GCT dataset. The establishment of a coherent global coastal classification is achieved through systematic rule-based criteria to identify landforms from high-resolution satellite imagery that define a coastal type. Seven major coastal categories have been identified; rocky, barrier island, strandplain, river mouth, estuary, tidal flats and vegetated regions (see Methods). To ensure these categories are mutually exclusive and collectively exhaustive, we first mapped the distribution of river mouths and estuaries, then assigned the dominant coastal environment by area within each 5 km stretch. Subsequently, we analyze flood exposure levels for the respective environments from combined rising sea levels and extreme storm surge events (Muis et al., 2020) for a CMIP6 projection using a high SSP5-8.5 emissions scenario from 1950 to 2050. This information is then matched against population estimates of the flooded areas (World Bank, 2023; Tatem, 2017) to highlight the potential inundation exposure to coastal communities.

## Methods

### Coastal environment mapping

#### Coastal classification scheme

There are numerous coastal classification schemes, including those by Johnson (1919), Valentin (1952), Cotton (1952) and others. Finkl (2004) provides a comprehensive review, highlighting that classification approaches vary significantly depending on their intended use. However, they can generally be categorized based on five key factors: (1) processes, (2) materials, (3) morphology, (4) developmental stage or age and (5) environmental context (Finkl, 2004). In the current study, we classify environments to link the information with morphologies and processes to better understand the future response of coastal types to sea-level rise. The GCT database is based on a modified version of the coastal sedimentary environment classification by Boyd et al. (1992). These coastal types include: (1) estuarine, (2) rocky, (3) river mouth, (4) barrier island, (5) vegetated, (6) strandplain and (7) tidal flats.

Our high-resolution mapping of dominant coastal environments at a 5-km scale ensures a consistent, reproducible and robust classification that effectively captures coastal variability at that given scale. Importantly, the GCT database does not incorporate hierarchical classifications (e.g., tidal flats within an estuary), as such classifications depend on mapping resolution. For instance, a traditional delta morphology can be defined at both a kilometer and hundreds-of-kilometer scale. Rather, the GCT classifications can be aggregated at larger scales, allowing hierarchical levels to capture internal variability in a coastal environment – for example, by linking the GCT to existing geospatial datasets, such as the global delta area database by Caldwell et al. (2019). Our classification scheme is thus similar to the three-class nonhierarchical classification of Mao et al. (2022) defining the dominant coastal environment within 2 km segments based on a machine learning approach.

The seven coastal types were selected based on the level of detail that can be discerned from satellite observations at the chosen 1:100,000 scale and 5 km segment size. As a result, the underlying substrate is not considered in our classification. Our primary goal is to delineate coastal segments that exhibit different responses to sea-level rise over short-term, decadal timescales (Boyd et al., 1992; Finkl, 2004). The classification process follows a two-step approach detailed below: first mapping river-influenced environments, then classifying the remaining coastline environments. The coastal environments each exhibit diverse responses to sea-level change. For example, barrier islands typically migrate landward as sea levels rise (Hoyt, 1967); many vegetated environments, such as mangroves, can keep pace with current sea-level changes (Woodroffe et al., 2016); while river mouths may submerge, depending on the balance between sediment supply, accommodation space and sea level rise (Nienhuis et al., 2023). When assessing future flood exposure, the 5-km scale serves as a potential link between global policy and regional to local adaptation strategies (Wolff et al., 2018). Notably, we distinguish between river mouth and estuarine environments, as described below, based on the characteristics such as a protruding river mouth or a funnel-shaped morphology. Our decision to classify river mouths rather than deltas reflects the need to account for along-strike variability in depositional environments within deltaic systems at different scales. This distinction is further detailed below and complemented by a flow diagram in the Supplementary Material.

### Coastal segmentation

We define the coastline based on the DINAS-COAST shoreline segments as outlined by Vafeidis et al. (2008). One benefit of this coastline representation is that it further allows the current database to integrate with existing DINAS tools for the socioeconomic assessments of coastal flooding for local, regional and global policy implementations. Additionally, this approach enables the classification of shoreline segments at a high 1:100,000 mapping scale within a feasible timeframe by initially dividing global coastlines into 5 km bins. This is achieved by manually assigning each segment to one of the seven classes according to a two-stage rule based on criteria within QGIS (2024), a geographical information system. The two-stage rule shown in Supplementary Figure S1 defines coastal morphologies based on the characteristics observed from satellite imagery and was followed to map each coastal type. Karst shorelines are classified as rocky and coastal environments

behind coral reefs are classified as barrier islands. In total, 691,000 km of coastline has been classified for 22,215 coastal segments.

The first step focused on identifying sedimentary environments along the coastline that have a river input. To constrain the database, we use the existing global distribution of deltas and rivers as defined by Caldwell et al. (2019) and the global geomorphic classification of beach, bedrock and wetland defined by Mao et al. (2022). Based on the degree of funneling at the river mouth, the presence or absence of a protrusion at the river mouth and/or high relief rocky versus low relief soft sediment topographic character, we define coastal environments with river input as either estuary, river mouth or rocky (Figure 1).

If a river connected to the coastline is not protruding or funnel-shaped, or if there is no river input, we define these sedimentary environments based on Step 2. This step involves classifying the remaining sedimentary environments including those coastal river inputs not defined by Step 1. For Step 2, the largest area of coastal morphological features belonging to an environment along a 5-km coastline segment was used to define the coastal type. Relatively narrow sedimentary islands that are parallel to the coastline are defined as barrier islands (Figure 1). An additional criterion to be defined as a barrier island is that the barrier island length has to be longer than the width of the lagoon and the barrier island width has to be less than twice the width of the lagoon. The regional/ continental scale classification of lagoonal systems of Dürr et al. (2011) has been used as a reference to constrain the location of barrier islands.

Coastal regions with low relief covered predominately by vegetation are classified as vegetated. This classification also includes agricultural regions along the coastline that are largely human-modified such as rice fields typically found in southeast Asia. High-relief regions that were not previously defined by Step 1, were included within the rocky subclass. Finally, strandplain and tidal flat regions were based on the dominant features within 1 km of landwards of the coastline, considering that several coastal environments may exist in this direction. If beach and chenier ridges were identified from satellite imagery, we defined the environment as strandplain (Figure 1). Here, the global sandy versus rocky coastline classification of Mao et al. (2022) was used as a reference to confirm our manual classification based on satellite imagery. Areas with blind channels, tidal flats and salt marshes are attributed to the tidal flat coastal type.

The accuracy of the GCT classification is based on a confusion-matrix comparison to 560 control observations (see Supplementary Figures S1 and S2) comparing the agreement between four mappers following the above workflow (e.g., Figure 1), which resulted in an overall accuracy of 82%. Vegetated and tidal regions are classified with the lowest agreement at 69% and 76%, respectively, whereas the highest agreement is for strandplains (90%) and rocky (88%) coastlines. River mouths, barrier islands and estuaries are classified at 81%, 85% and 85% agreement, respectively.

### Exposure data

#### Extreme Sea level inundation extent

By using the CMIP6 projection and an SSP5-8.5 emissions scenario, we map extreme sea-level inundation extent from the Global Tide and Surge Model indicators(Muis et al., 2020) to derive sea-level rise and extreme sea-level height based on the HadGEM3-GC31-HM climate model. Sea-level rise is measured at a mean annual highest high water (MHHW) line calculated against a 1986–2005 mean sea-level reference period for every decade between 1950 and 2050 at a 0.1° coastal grid point resolution. The return period is calculated for a 1-, 10-, 50- and 100-year event for the three climate reference periods of 1950–1981, 1985–2014 and 2021–2050. The MHHW and 12 return periods are each gridded at a resolution of 1,000 m based on the closest grid cell to any available coastal grid point.

To map the inundation height against the coastal topography, we utilize the global FabDEM dataset. This is a digital surface model at a resolution of 30 m that measures the surface of the earth (e.g., removed buildings, trees), providing improved global estimates of inundation extents along populated centers and vegetated ecosystems (Hawker et al., 2022). To compare each inundation extent to its coastal type, we assign each 30-m pixel to the closest coastline segment defined in the GCT database. While this model uses a simple inundation height against static topography without consideration of flood defenses, we do consider only the hydrologically connected exorheic component as defined by the HydroSHEDS dataset (Lehner et al., 2008). This approach is similar to many existing global storm surge risk models (Muis et al., 2016; Kulp and Strauss, 2019), considering that detailed flood protection data, including location and capacity, are often lacking or difficult to quantify on a global scale. As a reference, Nienhuis et al. (2022) estimate that roughly 26% of the world's population living on deltas may be protected by levees. Our maps thus provide insights into the potential exposure of flood inundation by coastal type, as well as highlight the value of existing nature-based and man-made coastal flood defense strategies (Slinger et al., 2021) by showing worst-case scenarios without mitigation strategies along the coastlines.

### Population estimates

To evaluate the flood exposure in coastal communities, we utilize a baseline 2020 population at an available 100 m grid resolution by the WorldPop project (Tatem, 2017). Historical and future projections are derived by applying decadal population change estimates by The World Bank from 1960 to 2050 to the 2020 baseline grid (World Bank, 2023). Population estimates by the World Bank are based on subnational census data including information on estimates of fertility, mortality and migration. To extend the population estimate to include 1950, we apply the same decadal population growth for the 1960s in order to have a corresponding length to the available extreme sea-level data. It is also important to note that historical and future projections do not contain internal migration patterns due to, for instance, famine, natural disasters, war or climate change (Kaczan and Orgill-Meyer, 2020). Nonetheless, this dataset provides the best available historical and future projections of populations at a resolution important to assess regional to local coastal inundation exposure. The population for each decade is subsequently compared to each coastal inundation level and coastal type.

### Results

### Global distribution of coastal environments

Our newly developed GCT dataset covers all coastal areas, excluding Antarctica. Here, we report the values within 10 km of the shoreline and less than 5 m in elevation (Figure 2). In summary, rocky shorelines are most prominent in northern high latitude regions covering an area of approximately 171,000 km$^2$ (17.4% of the total coastal area) and a length of 422,000 km (61% of the total coastal length). Strandplains are common globally, but a higher proportion are found along mid-latitude regions. In total, they

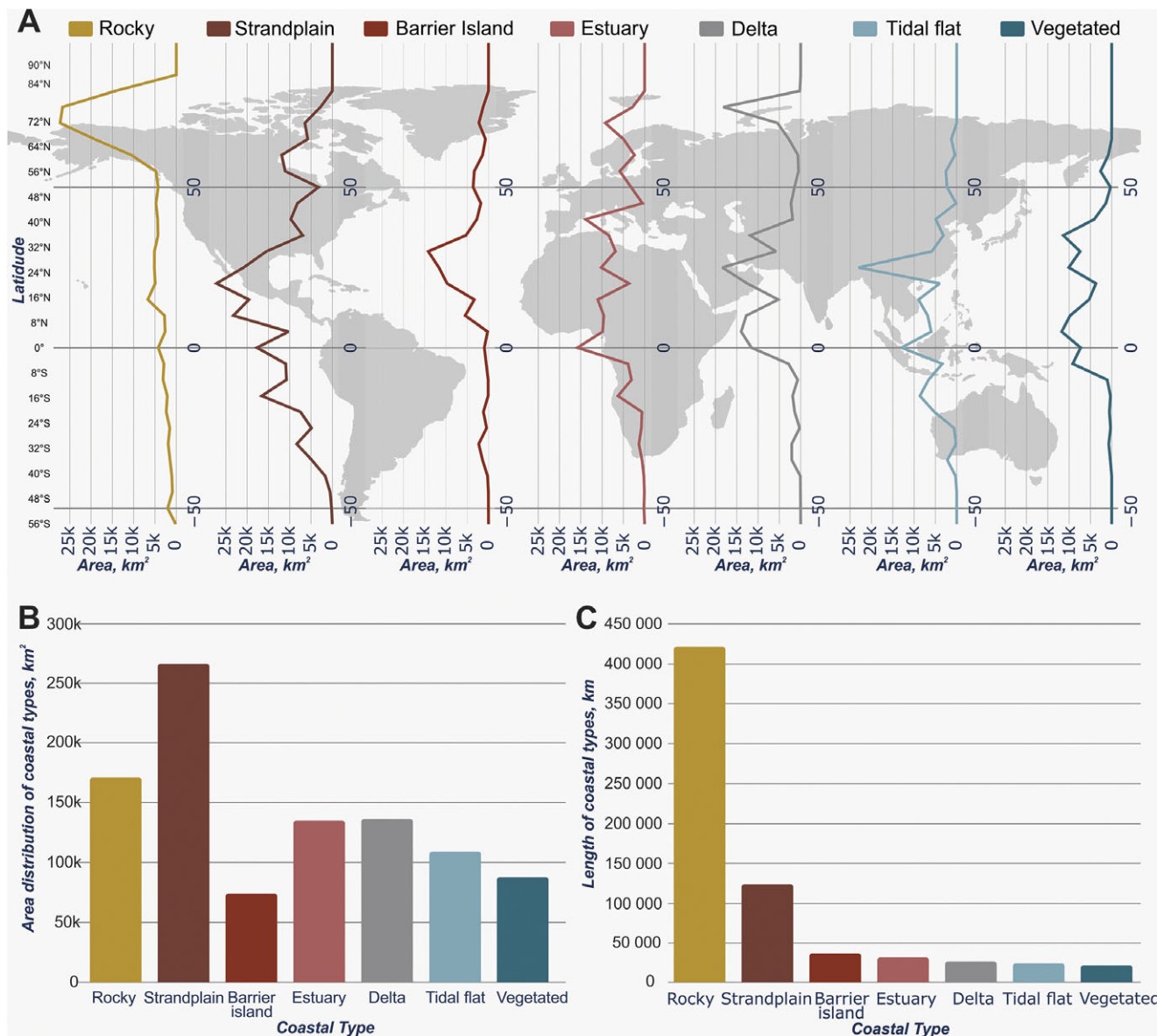

**Figure 2.** Global Coastal Type Distribution - A) distribution of coastal types in km2 binned into 10-degree latitude bin. B) Areal distribution of coastal types in km2. C) Length of coastal types in km.

cover the most area at 266,000 km$^2$ (27%) with a length of 123,000 km (18%). Barrier islands cover the least surface area at 74,000 km$^2$ (7.5%), yet lengthwise are third at 37,000 km (5.4%). These are found mostly in the northern hemisphere of mid-latitude coastlines. Globally, the distribution of estuaries is variable covering an area of 135,000 km$^2$ (14%) and a length of 32,000 km (4.7%). Similarly, river mouths show a variable geographic distribution, including a high-latitude peak. River mouths cover a total area of 136,000 km$^2$ (14%) and a length of 28,000 km (4%). Finally, tidal and vegetated environments are both found along the equatorial and mid-latitude regions covering an area of 109,000 km$^2$ (11%) and 88,000 km$^2$ (8.9%) and a length of 25,000 km (3.5%) and 23,000 km (3.3%), respectively.

### Present-day coastal flood exposure by coastal type

Based on the new GCT dataset, we compared each coastal type with the current population for present MHHW levels and for extreme sea-level events that combine MHHW and storm surge inundation

return periods of 1, 10, 50 and 100 years. The data reflect the number of individuals exposed to inundation without flood defense measures, a similar approach to many existing global coastal flood maps(Muis et al., 2016; Kulp and Strauss, 2019) (see Methods for more detail). Our results show that 44.9 million (24.6–83.4 at ±0.5 m) individuals lived below MHHW in 2020, with vegetated coasts (12.5 million), strandplains (9.4 million), estuaries (8.6 million) and river mouths (6.4 million) accounting for the highest numbers (Figure 3). In comparison, rocky and barrier island coastal communities are the least populated coastal types at present, representing 2.1 and 2.9 million, respectively, below the MHHW level.

For a 100-year extreme sea-level return period, more than 209 million people (152–277 at ±0.5 m) are exposed to coastal inundations. Coastal communities living along river mouth, vegetated and estuarine coastlines are the most populous, accounting for 61, 57 and 47 million people (Figure 3), respectively, or a combined 78% of the total population at exposed. The estuary and river mouth coastlines further show the highest increase in exposed populations when increasing extreme sea-level intensity from a 1- to 100-year

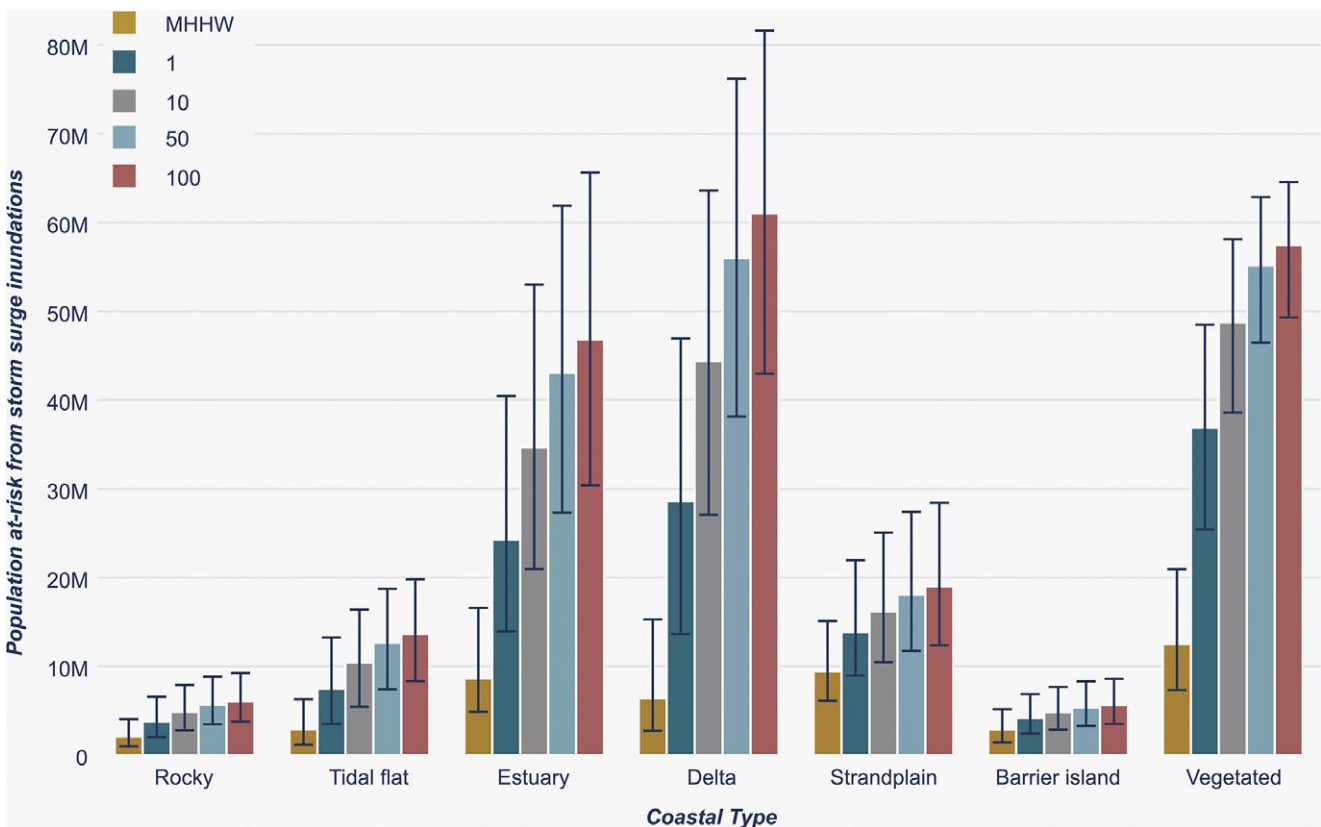

**Figure 3.** The 2020 population at-risk from extreme sea level inundations by coastal type. Scenarios include mean higher high water (MHHW) and 1, 10, 50 and 100 year flood return period with a +/- 0.5 m error bar in predicted MHHW or extreme sea level height.

return period, at a 92% and 113% increase, respectively. This finding reaffirms the well-documented notion that populations living on low-lying riverine systems are particularly vulnerable to storm surge levels (Eilander et al., 2020).

The impact of extreme sea-level inundations by coastal type will also vary regionally. As an example, for a 2020 10-year return period, a disproportionately high number of people affected (~131 million or 81%) live in Asia. Of these, most are living on or near river mouths or vegetated environments (Figure 4). In comparison, the less-affected continents of Europe, South America, Africa and Oceania have a higher proportion of communities living along strandplains. North America has a more variable distribution, with significantly populated coastal communities living along most of the different coastal types.

### Historical and future scenarios of extreme sea-level inundations

Historically (1950–2020), the number of people vulnerable to coastal flooding for a 1 in 10-year extreme sea-level event has increased by between 61% and 70% for most coastal types (Figure 5a), in part due to population growth, but also with a contribution from sea-level rise. The exceptions are rocky coasts showing less than a 52% increase from 1950 to the present-day. Future projections, accounting for population growth, sea-level rise and storm surge intensity change, show that populations on and in the vicinity of barrier islands will increase the most by 59% by 2050 for a 1 in a 10-year extreme sea-level event (Figure 5a). Strandplains are second with a 40% increase followed closely by tidal systems at 38%. Rocky, river mouth and

estuarine coasts will see a lower but still significant change ranging between a 16% and 26% increase, whereas vegetated regions will show a marginal change.

When examining the area that is potentially inundated by a 1 in 10-year return period extreme sea level, then rocky shorelines have reduced by nearly 5% in the inundated areas from 1950 to 2000 (Figure 5b). However, the area flooded along rocky coasts has since increased to the present-day and is expected to continue to increase by another 7% in the next three decades. The most vulnerable coastal types by inundated areas are vegetated environments showing a 12.5% increase since 1950 until today and another 8.7% expected to increase toward 2050. Strandplains show the smallest historical change with a maximum 6.2% increase since 1950, but this is projected to increase drastically, up to 12.3%, toward 2050. Finally, deltaic, estuarine, tidal and barrier island environments have observed a significant 7.4%–9.4% increase since 1950 till today and will add another 10.5%–12% by 2050.

### Discussion

Standardized classifications enable comparisons across diverse coastal settings, supporting localized decision-making while ensuring alignment with broader regional and global strategies (Wolff et al., 2018; Bongarts Lebbe et al., 2021). This structured approach helps prioritize adaptation efforts, guides sustainable coastal development and integrates nature-based solutions into climate resilience planning. Coastline environment classifications also play a crucial role in understanding and managing coastal ecosystems on a global scale. These classifications provide a structured framework for assessing

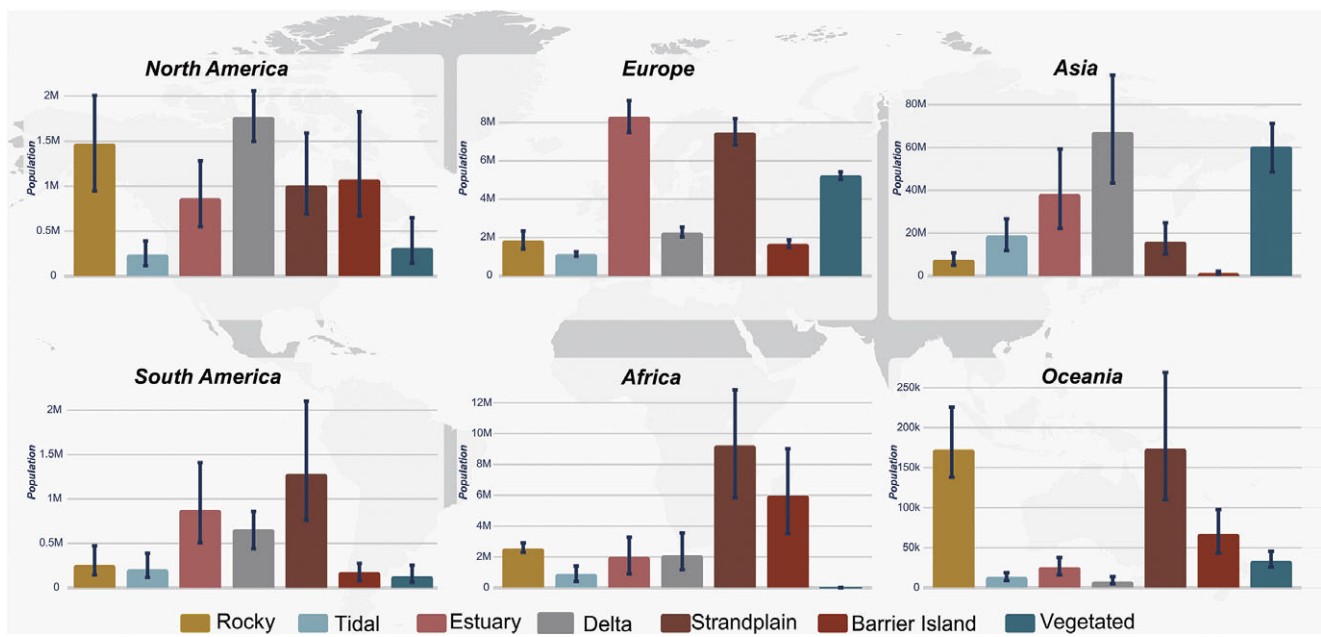

**Figure 4.** The 2020 population at-risk from a 10-year extreme sea level inundation by coastal type and continent with a +/- 0.5 m error bar in predicted flood height. See interactive map under the data availability section for more information.

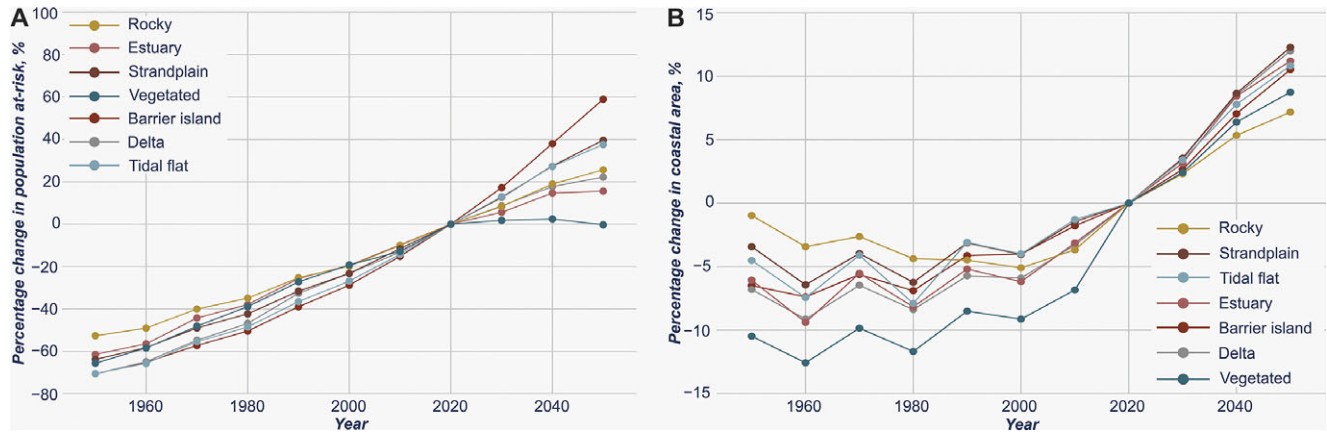

**Figure 5.** Historical records and future scenarios of coastal populations and area at risk from storm-surge inundations from 1950 to 2050 based on a SSP5-8.5 CMIP6 projection. A) Percentage change in population at risk from a 10-year storm-surge event by coastal type, relative to year 2020. B) Percentage change in coastal area inundated from a 10-year extreme sea level event by coastal type, relative to year 2020.

coastal vulnerability, resilience and exposure to environmental threats, including climate change and human activities (Sayre et al., 2019). Given that coastal ecosystems offer essential services – such as storm protection, carbon sequestration, biodiversity support and fisheries – accurate classifications help policymakers, researchers and stakeholders prioritize conservation and adaptation efforts. The GCT dataset presented in this manuscript helps to achieve that goal. Below, we have highlighted the implications of sea-level rise on the different coastal types and the causation for exposure trends.

### Impact of sea-level rise and extreme sea level on coastal environments

Barrier islands are typically formed under transgressive conditions of sea-level rise by vertical accretion concomitant with barrier island-lagoon couplet landward migration (Hoyt, 1967).

Strandplains, on the other hand, typically form under regressive conditions (Boyd et al., 1992) by adding new increments to the foreshore from longshore and cross-shore sediment sources, creating a seaward propagating topography with semi-parallel ridges and swales. Both barrier islands and strandplains, by creating coast-parallel ridges, often provide natural protection against storm surges. However, both are also sensitive to accelerated relative sea-level rise which are processes facilitating landward retreat of the shoreline (Mariotti and Hein, 2022). In particular, barrier islands provide important protection for large low-lying regions on their landward side.

Barrier islands and strandplains currently account for approximately 21 million people who are at risk from extreme sea-level events with a 10-year return period, representing only ~13% of the total global population exposed to such risks. While this figure is significantly lower than for river mouth and estuarine regions,

future projections indicate that barrier islands are among the fastest-growing population centers (Figure 5) (Tatem, 2017; World Bank, 2023). This trend is particularly pronounced in Sub-Saharan Africa (Figure 4), where rapid urbanization and population growth are expected to heighten exposure to coastal flooding. A key concern is the limited adaptive capacity of many Sub-Saharan African nations (World Bank, 2023), where lower income levels and constrained resources pose challenges for effective mitigation and resilience-building efforts (IPCC, 2023). As a result, coastal communities situated on or near barrier islands in this region are likely to experience heightened vulnerability to sea-level rise and extreme sea-level events (Kemgang Ghomsi et al., 2024). In contrast, strandplains – although also common in Africa – extend across other regions, such as Oceania, South America and Europe (Figure 4), where adaptation and mitigation capacities may be more robust.

Estuaries and tidal flats are currently occupied by 8.6 and 2.9 million people below MHHW, respectively (Figure 3). These environments are predicted to have, respectively, 20% and 40% increase in population exposed from 2020 to 2050 to a 10-year extreme sea-level return period (Figure 5a). These coastal regions (e.g., Southeast Asia, China, Indonesia, East coast of Africa) are typified by a mixed wetland and intertidal mudflat coastal environment and form typically under transgressive conditions by drowning of paleovalleys or under regressive conditions in tide-dominated deltas (Boyd et al., 1992). Depending on the rate of sea-level rise, the intertidal regions may either be eroded or inundated. Tidal currents will tend to propagate upstream, enhancing sediment mobility and flood dominance leads to higher sediment fluxes upstream. Depending on sediment availability, estuaries may either silt up or margins will be drowned, expanding the estuary (Leuven et al., 2019).

The influence of sea-level rise on river mouth environments will similarly depend on the interplay between sediment supply, vertical land movement (uplift and subsidence) and sea-level rise (Nienhuis et al., 2023). If the amount of accommodation produced by the combination of vertical land movement and sea-level rise outpace sediment supply, the low-lying river mouths will likely be drowned (Jervey, 1988; Nienhuis et al., 2023). Many of these coastal regions are affected by upstream dam-building (Zarfl et al., 2015), in addition to sand-mining (Hackney et al., 2021; Bendixen et al., 2023) and water pumping (Minderhoud et al., 2017) in the downstream coastal domain. Such activities will amplify the already devastating impact of storm surges and terrestrial flooding. We show that river mouths are the most populated (e.g., India, Bangladesh, Southeast Asia, Indonesia, Philippines, Egypt, West coast of Africa) sedimentary environment exposed to extreme sea-level events, with an estimated 6.4 million currently below MHHW and 44.4 million below a 10-year return period (Figure 3).

Vegetated coasts make up 8.9% of the world's coastal area, and host 12.6 million people below MHHW (Figure 3). Vegetated coasts (e.g., Florida, USA, Cuba, northern South America, parts of Indonesia) are efficient at trapping sediment and stabilizing the sea floor, which will reduce erosion due to sea-level rise. For instance, mangroves that can have growth rates up to 21 mm/year, are found across the range of different substrate environments. These are known to be resilient to moderately high rates of sea-level rise; however, human pressure is often the largest threat to this natural mitigation process (Woodroffe et al., 2016). From 2020 to 2050, there will be no major change in the population exposed along this type of coastline under a 10-year extreme sea-level return period (Figure 5a). However, the area of coastal inundation will increase by 8% in the same timeframe, suggesting that likely inundated regions are less populated. It is also important to note that a variety of flora exist across the range of coastal types, each with a different response

to sea-level rise and storm surges. While this is beyond the scope of the current work, the GCT dataset provides a baseline to further investigate these challenges.

Finally, rocky shorelines have their maximum distribution between 60 and 80°N (Figure 2a), mainly because of postglacial isostatic uplift (Milne and Shennan, 2013; Crosetto et al., 2020). Currently, 2.1 million people live below MHHW along rocky shorelines (Figure 3), with 6 million affected by a 100-year extreme sea-level return period. Pocket beaches between cliffs will be most susceptible to sea-level rise. Loose rocky coastlines, such as the limestone cliffs of the United Kingdom and France, will also likely experience accelerated erosion and shoreline retreat throughout this century (Dawson et al., 2009; Shadrick et al., 2022). Rocky coastlines, by their high gradient cross-shore topography, will be less affected by extreme sea-level events than the other coastal types.

### Causation of coastal inundation exposure

Population growth is the main cause for the historical and projected future increase in coastal inundation exposure (Supplementary Figure S3A). Here, we observe that for both an MHHW and a 10-year flood extreme sea-level event, coastal populations exposed to floods have increased by 45 and 138 million from 1950 to 2050, respectively. Sea-level rise and extreme sea-level change without population growth, in comparison, only contribute 5 and 13 million, respectively, to the increased population exposure. However, sea-level rise and extreme sea levels are becoming an increasingly important factor toward 2050 (Supplementary Figure S3B). This is in part not only due to a projected increase in the rate of sea-level rise (Fox-Kemper, 2021) and storm surge intensity (Muis et al., 2020), but also due to the observed and projected decline in the rate of population growth (World Bank, 2023). Crucially, it is important to recognize that future MHHW levels as a result of rising seas will dictate the extent of the landward storm surge floods.

The impact of sea-level rise on coastal inundation extent varies by coastal type over time (Supplementary Figure S3B). All coastal types, except rocky coastlines, show that MHHW is becoming an increasingly dominant factor in inundation compared to a 10-year extreme sea-level event from 1950 to 2050 (Supplementary Figure S4). By 2050, MHHW along barrier islands and strandplains will account for 65% and 54% of the total 10-year flood extent, respectively, while river mouth and vegetated regions contribute only 32% and 26%, respectively. This suggests that sea-level rise alone poses a more persistent threat to barrier islands and strandplains, whereas storm surges are more problematic for river mouths, vegetated and estuarine environments in terms of inundated areas (Figure 3). These distinctions are crucial, as the associated risks occur on different temporal and spatial scales and require distinct mitigation strategies. However, when examining the percentage change in the inundated area at annual MHHW levels since 1950, rising sea levels are also increasingly critical for river mouths and vegetated coasts, with projected increases of 50% and 37% by 2050, respectively (Supplementary Figure S3). Most of this increase is expected in the next three decades under the CMIP6 SSP5-8.5 emissions scenario.

Rocky shorelines show a similar contribution from MHHW to the maximum extent of a 10-year extreme sea-level event (Supplementary Figure S4), indicating that changes in MHHW extent have mirrored changes in exposure from storm surge flood extent. The absolute flooded area at the MHHW along the rocky coasts has not changed significantly from the 1950s until around 2010 (Supplementary Figure S3B), aligning with trends in 10-year

storm surge flood extent (Figure 5b). The reduction in storm surge flood extent from 1950 to approximately 2010 (Figure 5 and Supplementary Figure S3), despite an increase in the affected populations (Figure 5a), may be linked to postglacial isostatic uplift in sparsely populated high-latitude regions (Milne and Shennan, 2013). However, rising sea levels over the next three decades are expected to increase exposure along rocky shorelines by 2050.

### Study limitations and comparison

While the GCT database provides a globally consistent classification of coastal environments, it has certain limitations. Notably, it is limited to the resolution of observations from satellites at a 5-km scale, and does not provide a hierarchical or overlapping classification, which is common in many coastal classification schemes defined in the literature (Finkl, 2004). As a result, finer-scale features may be overlooked, which can be important for capturing subtle environmental variations. Additionally, a single coastal segment may encompass multiple environments with distinct processes and interactions, particularly in response to rising sea levels (Finkl, 2004; Nyberg and Howell, 2016). However, this effect is somewhat mitigated by our higher-resolution 5 km segmentation.

This study also employs a simple bathtub inundation model to assess coastal exposure across different environments. While this approach tends to overestimate inundation – since it compares extreme sea levels to static topography limited to watersheds connected to the ocean, as seen in previous studies (Muis et al., 2016; Kulp and Strauss, 2019) – its main advantage lies in its ability to analyze large datasets, including high-resolution topography, population distributions and emission scenarios. In contrast, while numerical hydrodynamic models such as CoSMoS (Barnard et al., 2019), SFINCS (Sebastian et al., 2021) and ADCIRC (Xie et al., 2016) have significantly improved spatial and temporal efficiency in recent years; however, the simulations remain constrained in their application at regional to global scales at high spatial resolution. Finally, it is important to acknowledge that the population estimates and projections used in this study do not account for migration dynamics or shifting population patterns due to climate change, including sea-level rise and extreme sea-level events. Nonetheless, our findings highlight how coastal exposure is changing across different regions and coastal types over time.

In comparison to other GCT classifications, Mao et al. (2022) also adopted a flat classification structure to identify 36.8% of the shoreline as wetland, 26.7% as beach and 36.5% as bedrock. In contrast, our study classifies only 11.5% of the shoreline as estuary, vegetated or tidal flats and 23.4% as strandplain or barrier islands. Additionally, we categorize 61% of the shoreline as rocky – significantly higher than Mao et al.'s 36.5% – a discrepancy likely due to their study region (60°N to 54°S), which excludes major rocky coastlines in Scandinavia, Russia, Canada and parts of the United States. Compared to deltaic areas, Edmonds et al. (2020) estimated a global deltaic area of approximately 847,000 $km^2$, whereas our more conservative estimate for river mouths is 136,000 $km^2$ (Figure 1). This difference reflects our more detailed and subdivided classification, which aims to capture along-strike variability in coastal environments at a fixed 5-km scale.

### Conclusions

A newly developed GCT dataset classifies the world's coasts into seven major categories: rocky, barrier islands, strandplains, river mouth, estuary, tidal flats and vegetated regions. This comprehensive database allows for an in-depth assessment of coastal flood exposure by type, revealing that by 2050, river mouths, estuaries and vegetated coasts are particularly exposed, with over 94 million people at risk from a 10-year extreme sea-level event. While these low-lying areas currently account for nearly 48% of the total exposed coastal population along 12% of the global coastline (~37% by land area), strandplains and barrier islands are projected to see a 40% increase in exposed populations. The findings emphasize the need for tailored risk mitigation strategies and future research that focus on the distinct vulnerabilities of each coastal environment to rising sea levels.

**Open peer review.** To view the open peer review materials for this article, please visit http://doi.org/10.1017/cft.2025.10001.

**Supplementary material.** The supplementary material for this article can be found at http://doi.org/10.1017/cft.2025.10001.

**Data availability statement.** The GCT database developed in this study has been deposited in the Zenodo database under accession code https://zenodo.org/records/14803853. An interactive map is available at https://bjornburrnyberg.users.earthengine.app/view/slr.

**Acknowledgments.** This project was supported by the Sea Level Projections and Reconstructions (SeaPR) project at the Bjerknes Centre for Climate Research. *This work is a contribution to IGCP Project 725 'Forecasting Coastal Change'.* The authors would like to thank the anonymous reviewer and Editor-in-Chief, Prof. Tom Spencer, for their valuable comments and constructive feedback, which significantly improved the manuscript.

**Author contribution.** BN designed the project and performed the data analysis. AG created the GCT database and prepared the manuscript figures. BN, AG, WHH, JN and JS discussed refined concepts and wrote and edited the manuscript.

**Competing interests.** The authors declare no conflict of interest.

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
