## [Reviewer Report]

Review of Manuscript: Flood Risk by Coastal Environment from 1950 to 2050 and its Impact on Coastal Communities<b></b>

General Comments

The manuscript provides an analysis of coastal exposure patterns by classifying coastal environments and linking these classifications to exposure data, namely flood extent, flood depth and population. While the study addresses an important topic, several issues hinder its overall clarity and impact. The title and claims throughout the manuscript are somewhat misleading, as the study focuses on exposure rather than risk, and the discussion of impacts on coastal communities is limited. A stronger focus on the purpose and implications of the classification, alongside transparent methodological choices, including the classification scheme and scale, would enhance the manuscripts impact. Key concepts such as vulnerability and the study’s innovation/research design are not clearly defined, and the discussion lacks depth in addressing the implications of findings, limitations, and comparisons with existing literature. Addressing these issues will significantly enhance the manuscript’s contribution to the field. Below, I outline specific feedback across major sections of the paper.

Title: The title is somewhat misleading. While it suggests an analysis of flood risk and impacts on coastal communities, the study focuses more on coastal exposure patterns. Risk involves a combination of exposure, vulnerability, and hazard, which is not fully addressed. A revised title could better reflect the scope of the work.

Impact Statement: The statement claims to identify “critical vulnerability patterns,” but the classification mainly pertains to exposure, not vulnerability. This should be rephrased to avoid overstating the study’s contributions.

Introduction:

Line 51: I would suggest to replace “predicted coastal inundation” with “projected” or “anticipated”

Line 53-54: The statement that coastal environment types are critical for assessing risks is true but may hold more validity on a local scale rather than a global one. Clarification of how the study could improve our understanding here would improve accuracy/position the work implication better.

Line 3-6 (Page 2): The claim that “most studies focus on deltas” is a bit overstated. While deltas are well-studied, other environments have been explored extensively too. Replacing “most” with “several” would improve precision.

Line 9-12 (Page 2): The authors claim a lack of global coastal type classifications/assessments (which is correct), but relevant studies exist. Citing and contrasting these studies would better position the work within the existing literature.

Some useful references:

1. Athanasiou, Panagiotis, Ap Van Dongeren, Maarten Pronk, Alessio Giardino, Michalis Vousdoukas, und Roshanka Ranasinghe. „Global Coastal Characteristics (GCC): A Global Dataset of Geophysical, Hydrodynamic, and Socioeconomic Coastal Indicators“. Earth System Science Data 16, Nr. 7 (29. Juli 2024): 3433–52. https://doi.org/10.5194/essd-16-3433-2024.

2. Mao, Yongjing, Daniel L. Harris, Zunyi Xie, und Stuart Phinn. „Global Coastal Geomorphology – Integrating Earth Observation and Geospatial Data“. Remote Sensing of Environment 278 (September 2022): 113082. https://doi.org/10.1016/j.rse.2022.113082.

3. Luijendijk, A., Hagenaars, G., Ranasinghe, R. et al. The State of the World’s Beaches. Sci Rep 8, 6641 (2018). https://doi.org/10.1038/s41598-018-24630-6

4. Vafeidis, A. T. et al. A New Global Coastal Database for Impact and Vulnerability Analysis to Sea-Level Rise. J COASTAL RES 244, 917–924, https://doi.org/10.2112/06-0725.1 (2008).

5. Wolff, C., Vafeidis, A., Muis, S. et al. A Mediterranean coastal database for assessing the impacts of sea-level rise and associated hazards. Sci Data 5, 180044 (2018). https://doi.org/10.1038/sdata.2018.44

The authors classify estuaries and river mouth: What is the difference? Would be deltas and estuaries more precise?

Coastal Classification: The rationale for the seven categories needs justification. For example, why distinguish river mouths and estuaries instead of combining them? Similarly, vegetated coasts often overlap with other types (e.g., mangroves in estuaries in a sandy environment). Providing a clear reasoning for the classification choices is essential.

Methods

Line 51 (Page 2): The manuscript states that the classification helps “better understand the future response of coastal types to sea level rise,” but the link between classification and response is not well-established in the manuscript. This requires elaboration.

Scale: A 5 km segmentation may fail to represent fine-scale coastal variability, which might be fine for a global application. However, please justify why this unit/length was chosen and discuss potential implications for the results.

Hierarchical classification (Page 3, Line: 4-9): The scope of the study needs to be better justified (the purpose of the classification). Additionally, the authors write that the mapping intends to classify the coast based on the responses to SLR on a short-term decadal scale (Page 3 Line 7: “Our intent is to map the environment-segments that will respond differently to sea level rise on the short-term decadal scale.”)– what response variability are you expecting (between the different coastal classes)? Why did you develop the 7 classes and why are you not using a hierarchical classification?

The DINAS_Coast coastline: The coastline is pretty coarse. While I do understand the advantages of being able to use the DIVA database – it is not really relevant to your research question/design as you not using any data related to the database. There are much better coastlines – a better justification would be useful.

Data Sources: The datasets used for segmentation (e.g., sedimentary environments) and criteria for classification are not clearly outlined/referenced. Including e.g. a table summarizing datasets, their sources, and purposes would greatly enhance transparency.

Vegetation Class: Vegetation often interacts with underlying coastal materials or grows on specific coastal material(e.g., sand, rock) or behind it. How does the classification account for this interaction? Some vegetated environments (e.g., salt marshes) benefit from inundation, challenging the assumption that all inundated vegetated areas are vulnerable.

Classification Methodology: The manual classification approach requires a detailed description. What were the steps, criteria, and tools used for satellite image interpretation? Validation methods should also be clarified, including the data source for the 560 control observations. Referencing studies like Buscombe et al. (2023) could provide additional validation insights.

- Reference: Buscombe, D., Wernette, P., Fitzpatrick, S. et al. A 1.2 Billion Pixel Human-Labeled Dataset for Data-Driven Classification of Coastal Environments. Sci Data 10, 46 (2023). https://doi.org/10.1038/s41597-023-01929-2

Exposure Data: The distinction between exposure data and coastline segmentation needs clearer justification. For instance a section header clearly separating the two could help.

Floodplain Connectivity: Some flooded areas (e.g., Florida example – please see my screen shot - not able to insert it here but happy to provide it, if needed) seem disconnected from the ocean, which could lead to overestimations of exposure. This methodological choice should be addressed. It is very hard from the manuscript to know what methodological choices have been made.

Results

Visualization: Figures are a strong point of the paper, especially Figure 1.

Data viewer: The addition of coastal classifications in the data viewer would further enhance accessibility.

Exposure vs. Vulnerability: Results focus on exposure, yet conclusions often imply vulnerability. This distinction should be consistently maintained.

Discussion

Vulnerability Claims: Statements like “the most vulnerable coastal type by inundated area are vegetated environments” (Page 8, Line 48) need stronger justification. Some vegetated environments, like salt marshes, are naturally adapted to inundation and may not align with conventional notions of vulnerability.

Adaptation Implications: The discussion should connect findings to specific adaptation strategies for different coastal types. For instance, how should adaptation approaches differ between rocky coasts and vegetated environments?

Study Limitations: A more thorough discussion of limitations is needed: The static bathtub model used for flooding is simplistic. Mention/discuss alternatives (e.g., reduced-complexity models) and their potential advantages; or the limitations of the results in general. For instance, coarse resolution of the segmentation and dataset limitations should also be highlighted. Or challenges in classification (e.g., overlapping categories) should be acknowledged.

Comparative Analysis: Comparing findings with other global studies would strengthen the discussion and highlight the study’s contributions.

Results part (Page 9 Line 57 – Page 10 Line 29): This part of the discussion reads more like results, and it might be good to restructure it.

---

## [Editor Report]

The reviewer has identified a number of points that would need to be addressed for improving manuscript. I would recommend the authors to carefully address all the points that the reviewer has raised.

Paricular attention should be given to the use of terminology (e.g. risk, exposure, vulnerability) as this is currently not consistent (and sometime incorrect). Also, the authors should place their study in the context of other coastal classifications in order to clarify the novelty of their study.

With respect to the use of the bathtub method for the calculation of the potential floodplain, the authors should discuss the advantages and limitations of the method, also in comparison with new methods employed for this purpose.

---

## [Reviewer Report]

Thank you for your thorough revisions and the substantial improvements made to the manuscript. I appreciate the effort put into refining the presentation of the methodology, which have significantly enhanced the clarity and readability of the study. Congratulations on this progress!

Before the paper is ready for publication, I have a few remaining comments and suggestions that I believe would further strengthen the manuscript:

•Related to A7: You mention that the definition of river mouths differs from deltas because your focus is on sedimentary environments that will change on a decadal scale rather than over longer time scales. However, this distinction is not fully reflected in the text. It might be beneficial to explicitly state the primary goal of the classification, as I believe this has been improved substantially but could still be clarified further. From my reading, the classification appears to aim primarily at capturing river input and how they and other coastal types respond on a decadal timescale. However, the manuscript seems to frame its purpose slightly differently in different sections (one example e.g. Line 98: “by mapping the dominant environment at a 5km resolution”). It may be helpful to ensure this alignment is clear.

• A9: I would argue that using the different responses of each classification type to justify the classification itself makes sense (maybe somewhere between line 141-153 track change version). Since one of the main purposes stated in the manuscript is to reflect decadal responses to sea-level rise (if I understand correctly), I suggest briefly explaining how the response varies across classification types and citing relevant literature to support this.

• A10: A question regarding mixed types—at a 5 km scale, it is likely that multiple types are represented within a segment (Estuary, barrier island and vegetated). From my understanding of the manuscript, the primary category always be determined by step 1 (if a river mouth exists), right? Clarifying how these mixed cases are handled in the classification could be useful (you are mentioning briefly that this is a limitation, but I think it would be good to highlight in the methods section). And elaborating on the effects of the classification in the limitations section would be beneficial for the manuscript (adding 1-2 additional sentences to the current discussion in the section ‘Study limitations and comparison’).

• A11: The methodology and overall structure suggest that the primary aim of the classification is to identify river mouths and estuaries (river input), which is a valuable and logical approach. However, this intent is not always as clearly conveyed in the text. It may be worth reconsidering how the paper is positioned to ensure that the main objective is immediately clear to readers.

• A15: I suggest moving the decision tree figure into the main text rather than the supplementary material. This figure provides essential context regarding the methodological choices in the classification and would be more useful within the primary discussion.

• A17: In the methods section, you state that only hydrologically connected exorheic components are considered. However, from the Florida example (which I unfortunately cannot attach here but have sent to the editor), it appears that non-connected floodplains may be included. This may be worth reviewing in the results section to ensure consistency and minimize potential errors.

Overall, I believe these refinements will help to further clarify the manuscript and ensure that the classification framework is well understood. I look forward to seeing the final version.

---

## [Editor Report]

I would like to thank the authors for carefully considering and addressing in detail all the comments and suggestions of the reviewer and the editors. I believe that the manuscript has improved considerably in terms of clarity and accuracy during the review process and can now be accepted for publication.